# Effect of Six Insecticides on Egg Hatching and Larval Mortality of *Trogoderma granarium* Everts (Coleoptera: Dermestidae)

**DOI:** 10.3390/insects11050263

**Published:** 2020-04-25

**Authors:** Maria C. Boukouvala, Nickolas G. Kavallieratos

**Affiliations:** Laboratory of Agricultural Zoology and Entomology, Department of Crop Science, Agricultural University of Athens, 75 Iera Odos str., 11855 Athens, Attica, Greece; nick_kaval@aua.gr

**Keywords:** concrete, contact insecticides, eggs, larvae, khapra beetle, food

## Abstract

The khapra beetle, *Trogoderma granarium* Everts (Coleoptera: Dermestidae), is one of the most destructive insect species of stored food worldwide and is subjected to strict phytosanitary legislations. In the present study, we evaluated the egg hatching and larval mortality of *T. granarium* on concrete surfaces treated with six insecticides (i.e., α-cypermethrin, chlorfenapyr, deltamethrin, pirimiphos-methyl, pyriproxyfen, and *s*-methoprene) that are registered in Greece for surface treatment and exhibit a broad spectrum of different modes of action. Furthermore, we investigated the influence of the presence of food on egg hatching and larval mortality. Egg hatchability on treated concrete was higher in tests with the presence of food for all tested insecticides, with the exception of *s*-methoprene. In contrast, larval mortality was lower in treatments where there was nourishment for all insecticides. No egg hatching was recorded on concrete treated with pirimiphos-methyl where there was no food, while with the addition of food, the egg hatching did not exceeded 26.7% after 6 days of exposure. The highest percentage of hatched eggs was recorded on concrete treated with chlorfenapyr (87.7% with food vs. 76.7% without food), followed by deltamethrin (76.7% with food vs. 63.3% without food), pyriproxyfen (50.0% with food vs. 42.2% without food), and α-cypermethrin (28.9% with food vs. 6.7% without food). In the case of *s*-methoprene, more eggs were hatched in the absence of food (91.1%) in contrast to in the presence of food (86.7%). Regarding mortality, all larvae were dead after 5 days of exposure on pirimiphos-methyl-treated concrete with food. Furthermore, larvae died faster in treatments without food. For α-cypermethrin, 100% mortality was recorded after 4 days of exposure, while with presence of food, all larvae died after 6 days. Chlorfenapyr caused complete mortality of larvae after 5 days of exposure on concrete without food and after 8 days with food. In the case of deltamethrin, 100% mortality was recorded after 7 days in the absence of food and 8 days in the presence of food. Regarding pyriproxyfen, complete mortality was not recorded when food was present, reaching 94.1% 14 days postexposure. However, after 12 days, all larvae died in treatments without food. Although egg hatching was higher in the case of *s*-methoprene on concrete without food, larval mortality was 100% after 8 days of exposure. Nevertheless, when there was food, 87.3% of the exposed larvae died after 13 days. Therefore, it becomes evident that sanitation of storage facilities before the application of contact insecticides is a key factor for the successful control of *T. granarium* in the egg stage.

## 1. Introduction

The khapra beetle, *Trogoderma granarium* Everts (Copleoptera: Dermestidae), is a serious stored-product insect pest of economic importance that is subjected to strict phytosanitary measures in several regions worldwide [1,2,3,4,5]. It affects the quality and quantity of various stored products causing losses [6,7], but can also be dangerous to public health by contaminating foodstuffs with body parts [8,9]. This species is able to feed upon a wide spectrum of commodities of plant and animal origin; however, it shows greater preference towards grains and starch products [6,10,11,12,13,14,15,16,17]. Recently, Kavallieratos et al. [18] showed that *T. granarium* rapidly reproduces on animal products, herbs, nuts, and pulses. Its development takes place at temperatures between 18.44 and 40 °C with an optimum at 34.52 °C [17], and it can easily compete other stored-product insects, such as the lesser grain borer, *Rhyzopertha dominica* (F.) (Coleoptera: Bostrychidae), and the rice weevil, *Sitophilus oryzae* (L.) (Coleoptera: Curculionidae) [6]. According to Kavallieratos et al. [6], 30 and 35 °C are the favorable temperatures for the establishment of high larval populations of *T. granarium* either alone or in combination with *R. dominica* and *S. oryzae* even after 65 days. Under biotic and abiotic conditions that are unfavorable for the development of larvae (e.g., temperature < 30 °C, overpopulation, lack of food), larvae may pass facultative diapauses for up to 8 years [19,20]. Diapausing larvae are particularly tolerant to insecticidal treatments [21,22,23] and nonchemical methods, such as elevated temperatures [24]. It is also well documented that nondiapausing larvae are not easily controlled in stored cereals, especially in maize and rough maize [25], on different types of surfaces, such as concrete [26,27,28] or various types of packaging materials [29]. 

Several studies have shown that *T. granarium* larvae are more tolerant than adults to different contact insecticides when they are applied as grain protectants [25] or as surface treatments [26,27,28,29,30]. For instance, Kavallieratos et al. [25] reported that among cypermethrin, deltamethrin, silicoSec, *s*-methoprene, and spinosad, only pirimiphos-methyl caused 100% larval mortality in wheat, barley, maize, and rough rice treated with the double label dose 7 days postexposure. Recently, the application of 1000 ppm of the furanosesquiterpene isofuranodiene on stored wheat led to 96.7% mortality of the exposed *T. granarium* adults after 7 days post-treatment, while larval mortality did not exceed 37.8% [31]. Previous studies have also documented that contact insecticides with different modes of action provide different levels of efficacy, as surface treatments, against larvae and adults of *T. granarium* [26,27,29]. However, limited information about the control of *T. granarium* eggs is available. The only known study that deals with this issue was conducted by Ali et al. [32] who examined the ovicidal effect of seven insect growth regulators (IGRs), i.e., lufenuron, flufenoxuron, pyriproxyfen, tebufenozide, methoxyfenozide, triflumuron, and buprofezin, against *T. granarium* eggs in glass vials. Therefore, the objective of the present study was to investigate the impact of six registered insecticides in Greece, that cover an extensive spectrum of different modes of action, i.e., the organophosphate pirimiphos-methyl, the pyrethroids deltamethrin and α-cypermethrin, the pyrrole derivative chlorfenapyr, and the IGR juvenile hormone analogues (JHAs) pyriproxyfen and *s*-methoprene, as concrete treatments for egg hatching and larval mortality of *T. granarium* when food was present or absent.

## 2. Materials and Methods

### 2.1. Insects, Commodity, and Insecticides

*T. granarium* eggs were taken from a colony maintained at the Laboratory of Agricultural Zoology and Entomology, Agricultural University of Athens. This insect colony was established in 2014 from insects collected in Greek storage facilities and reared on wheat at 30 °C, 65% relative humidity, in continuous darkness. White soft wheat flour (clean, without infestation, and untreated with pesticide) (a variety mixture, made from the endosperm only) was used in the bioassays, in the cases where food was provided. The following six insecticidal formulations were used in the experiments: Power SC with 62.4 g/L α-cypermethrin active ingredient (a.i.) (provided by Hybrid Hellas, Metamorphossis, Greece), Phantom EC with 21.45% chlorfenapyr a.i. (provided by BASF Hellas, Amaroussion, Greece), K-Othrine WG with 25% deltamethrin a.i. (provided by Bayer Hellas, Amaroussion Greece), Actellic EC with 50% pirimiphos-methyl a.i. (provided by Syngenta, Anthousa, Greece), Admiral EC with 10% pyriproxyfen a.i. (provided by Hellafarm, Amaroussion, Greece), and Biopren BM EC with 19% *s*-methoprene a.i. (provided by Farma-Chem SA, Thessalokini, Greece).

### 2.2. Bioassays

The insecticidal formulations were applied at the labeled doses for surface treatments. Thus, α-cypermethrin, chlorfenapyr, deltamethrin, pirimiphos-methyl, pyriproxyfen, and *s*-methoprene were examined at 0.10 mg a.i/cm^2^, 0.11 mg a.i/cm^2^, 0.005 a.i/cm^2^, 0.05 a.i/cm^2^, 0.00023 a.i/cm^2^, and 0.00030 a.i/cm^2^, respectively. The experiments were conducted in a completely randomized block design, with three subreplicates and three replicates in Petri dishes (8 cm diameter by 1.5 cm high) with a surface area of 50.27 cm^2^ each. Twenty-four hours before the beginning of the tests, the bottoms of the dishes were covered with the CEM I 52.5 N material (Durostick, Aspropyrgos, Greece) to make the concrete surface. The upper internal parts of all dishes were coated by polytetrafluoroethylene (60 wt % dispersion in water) (Sigma-Aldrich Chemie GmbH, Taufkirchen, Germany) to prevent the escape of the exposed larvae. The concrete surface of individual dishes was sprayed with 1 mL of an aqueous solution, as a fine mist, that contained the appropriate volume of α-cypermethrin, chlorfenapyr, deltamethrin, pirimiphos-methyl, pyriproxyfen, or *s*-methoprene corresponding to each labeled dose. Spraying was carried out using an AG-4 airbrush (Mecafer S.A., Valence, France). After spraying with each formulation, the airbrush was cleaned with acetone and then the next formulation was applied on individual dishes. Half of the dishes contained food, 0.5 g of white soft wheat flour, which was sprinkled over the surface area of the concrete 1 day after spraying. Before tests started the moisture content of the flour was adjusted to 13.5 ± 0.5% as determined by a moisture meter (mini GAC plus, Dickey-John Europe S.A.S., Colombes, France) either dried inside oven at 50 °C or hydrated with distilled water according to its initial moisture content [16,33]. An additional series of dishes with concrete surfaces were prepared and sprayed with distilled water, with a different AG-4 airbrush, as described above, to serve as controls.

To obtain eggs for experimentation, 200 unsexed adults, approximately 7 days old, were transferred from the culture to a 250 mL glass jar that contained 125 g white soft wheat flour for 1 day [17]. Then, the separation of adults and eggs from the flour was conducted with a No. 20 and a No. 60 USA standard testing sieve (Advantech Manufacturing, Inc., New Berlin, WI). The eggs that remained on the mesh openings of the sieve were put in a Petri dish and kept for 3 days before the beginning of the experiments, at 30 °C and 65% relative humidity. According to Kavallieratos et al. [34], the development of eggs ranged between 4.68 and 4.81 days at 32.5 °C and 65% relative humidity. At the fourth day, three series of 10 eggs each were transferred carefully with a slender haired brush (Cotman 111 No 000, Winsor and Newton, London, UK) to ten dishes that contained sprayed concrete (one egg per dish, i.e., 10 dishes per insecticide or control per food treatment for each series of eggs). All lids of the dishes had a circular aperture covered by mousseline (a perforated material) to allow for adequate airing inside the dishes. Then, all dishes were transferred into incubators set at 30 °C and 65% relative humidity, in continuous darkness. The egg hatching and mortality of the emerged larvae of each dish were observed daily for 14 days. Mortality was determined by nudging larvae gently with a brush (Cotman 111 No 000, Winsor and Newton, London, UK) to detect any response under an Olympus stereomicroscope (Olympus SZX9, Bacacos S.A., Athens, Greece). Different brushes were used per insecticidal formulation and controls.

### 2.3. Data Analyses

Data were analyzed separately for the hatched eggs and larval mortality with the presence or absence of food by following the repeated measures model [35]. The main effect was the insecticide, while the response variables were egg hatching or larval mortality. The repeated factor was exposure. Prior to analysis, the percentages of egg hatching or larval mortality were arcsine square root-transformed to normalize variance [36,37]. All analyses were conducted using the JMP 14 software [38]. Means were separated by the Tukey–Kramer honest significant difference (HSD) test at 0.05 probability [39]. 

## 3. Results

### 3.1. Egg Hatching

All main effects and the associated interactions were significant with or without food (Table 1). When food was present, after 2 days of exposure, no egg hatching was recorded in all treatments, even in the control dishes (Table 2). Three days later, 24.8% of the eggs were hatched on concrete treated with pyriproxyfen, while in all other treatments, hatching was lower and ranged between 3.3% and 23.3%. In the control dishes, a significantly higher egg-hatching rate was recorded (81.1%). After 5 days of exposure, the proportion of hatched eggs increased in all tested insecticides reaching a maximum value of 78.9% on *s*-methoprene-treated concrete. However, egg hatchability remained low in the treatments with α-cypermethrin and pirimiphos-methyl, where the egg hatchability did not exceed 22.2% and 24.4%, respectively. All exposed eggs (100%) in the untreated concrete were hatched. The highest proportion of egg hatching for α-cypermethrin (28.9%) and pirimiphos-methyl (26.7%) was recorded 6 days postexposure. The highest number of eggs hatched on concrete treated with deltamethrin (76.7%), chrorfenapyr (87.7%), *s*-methoprene (86.7%), and pyriproxyfen (50.0%) noted after 7, 8, 9, and 11 days of exposure, respectively.

In most cases, the absence of food affected the hatchability of eggs. On concrete treated with pirimiphos-methyl, no egg hatching was recorded. In treatments with α-cypermethrin, egg hatching remained extremely low and did not exceed 6.7%, after 5 days of exposure. For chlorfenapyr, deltamethrin, pyriproxyfen, and *s*-methoprene, the hatchability of eggs was higher 3 days postexposure in relation to the presence of food, i.e., 31.1%, 24.4%, 34.4%, and 48.9%, respectively. However, the maximum values of hatched eggs were higher when food was absent, i.e., 76.7% for chlorfenapyr, 63.3% for deltamethrin, and 42.2% for pyriproxyfen, 6, 6, and 5 days postexposure, respectively. In treatments with *s*-methoprene, the above trend was reversed since the maximum proportion of hatched eggs (91%) was observed when food was present after 5 days of exposure.

### 3.2. Larval Mortality

In the treatments with food, 90, 26, 79, 69, 24, 45, and 78 larvae were examined corresponding to controls, α-cypermethrin, chlorfenapyr, deltamethrin, pirimiphos-methyl, pyriproxyfen, and *s*-methoprene, respectively. In treatments without food, 89, 6, 69, 57, 0, 38 and 82 larvae were examined corresponding to controls, α-cypermethrin, chlorfenapyr, deltamethrin, pirimiphos-methyl, pyriproxyfen, and *s*-methoprene, respectively. All main effects and the associated interactions were significant between and within the exposure intervals (Table 1). In the control dishes, no larval mortality was recorded with or without food (Table 3). Regarding the tested insecticides, the presence or absence of food affected their insecticidal activity. For instance, after 1 day of exposure, no larval mortality was recorded in all treatments that contained food, while in the absence of food, 3.6% and 5.7% of the newly emerged larvae died on concrete treated with chlorfenapyr and deltamethrin, respectively. In treatments with food, 100% mortality of *T. granarium* larvae was noted after 5 days of exposure to pirimiphos-methyl, followed by α-cypermethrin after 6 days, and chlorfenapyr or deltamethrin after 8 days. When food was absent, 100% mortality was noted 4, 5, and 7 days postexposure on concrete treated with α-cypermethrin, chlorfenapyr, and deltamethrin, respectively. For treatments with pyriproxyfen and *s*-methoprene, 100% mortality was recorded after 12 and 8 days of exposure, respectively, without food. When food was available, maximally 94.1% and 87.3% of the larvae exposed to pyriproxyfen and *s*-methoprene were found dead, respectively.

## 4. Discussion

The findings of the current study indicate that exposure of *T. granarium* eggs to the tested insecticides may slow the spread of this species since they reduced egg hatching and eventually killed almost all emerged larvae in several cases. According to our results, pirimiphos-methyl and α-cypermethrin were by far the most effective insecticides against *T. granarium*. Furthermore, the presence or absence of food on the treated surface regulated the activity of the insecticides with the exception of *s*-methoprene. In general, the highest levels of egg hatching were observed in treatments where food was present, indicating that an amount of each insecticide was absorbed and/or degraded by flour, and/or flour acted as a protective layer between the eggs and toxicants leading to a reduction in their effectiveness. In previous studies, it has been shown that aerosols (e.g., pyrethrins mixed with methoprene or pyriproxyfen) [40], pyriproxyfen [41], or chlorfenapyr [42] may be partially absorbed by flour on treated concrete surfaces. Furthermore, Arthur [43] reported that flour prevented adults of the red flour beetle, *Tribolium castaneum* (Herbst) (Coleoptera: Tenebrionidae), from touching the concrete surface treated with cyfluthrin, an issue that led to increased survival of the exposed individuals. These are reliable scenarios taking into account that eggs are not a mobile biological stage, so they could avoid the exposure to insecticides on concrete only indirectly. Similarly, Toews et al. [44] showed that the mortality of eggs of *T. castaneum* decreased linearly with the increase in the quantity of flour treated with pyrethrins or esfenvalerate. 

Although pyriproxyfen and *s*-methoprene responded differently in the presence and absence of food, the former exhibited a higher overall toxicity to eggs than the latter. Previous studies have documented the variable effect of these IGRs on eggs in other stored-product pests. For example, pyriproxyfen showed very low ovicidal activity to the eggs of the cowpea beetle, *Callosobruchus maculatus* (F.) (Coleoptera: Bruchidae), in treated *Vigna unguiculata* (L.) Walp. (Fabales: Fabaceae) (LC_50_ = 91.9 mg/kg seeds) [45]. The application of *s*-methoprene on filter paper at doses ranging between 0.00003 and 0.3 led to 62.5–26.7% hatching of eggs of *R. dominica*, 7 days postexposure [46]. Recently, Ali et al. [32] examined the ovicidal effect of seven IGRs, including pyriproxyfen, against *T. granarium* eggs on glass surfaces. The authors reported that pyriproxyfen allowed 38.83% egg hatching, which was the lowest percentage among the tested IGRs. Nevertheless, in our experiments, by applying the same IGR on concrete, we found that 42.2% of the exposed eggs were hatched, indicating that concrete, as a porous material, can absorb a greater quantity of the insecticide compared to nonporous materials such as glass [47], leading to higher rates of egg hatching. Similarly, Arthur et al. [48] reported that no adult emergence was recorded when young larvae of *T. granarium* were exposed on a nonporous metal surface treated with pyriproxyfen, while on the porous treated concrete surface, adult emergence was almost 20%.

Despite the fact that deltamethrin is a pyrethroid insecticide, it was less toxic to eggs than the pyrethroid α-cypermethrin either in the presence or the absence of food. The moderate toxicity of deltamethrin has also been recorded for eggs of the diamondback moth, *Plutella xylostella* (L.) (Lepidoptera: Plutellidae), given that the hatched eggs did not exceed 38.4%, 6 days postexposure on *Brassica rapa* L. ssp. *chinensis* (L.) Hanelt (Brassicales: Brassicaceae) at 12.5 g a.i./ha [49]. This issue may be attributed to the low sensitivity of the nervous system of the embryo to the neurotoxic deltamethrin [49]. However, on the basis of our results and previous studies, different neurotoxic pyrethroid or organothiophosphate insecticides, such as α-cypermethrin or pirimiphos-methyl, respectively, exhibit elevated ovicidal toxicity. For instance, pirimiphos-methyl achieved a high ovicidal toxicity for *C. maculatus* (LC_50_ = 1.82 mg/kg seeds) [45]. The non-neurotoxic pyrrole chlorfenapyr [50,51,52] exhibited a low ovicidal effect, with or without food, but it is a species dependent phenomenon given that this compound induced a high (87%) ovicidal activity for exposed *T. castaneum* larvae [53]. Further experimentation is needed to clarify these issues. 

Although α-cypermethrin, chlorfenapyr, deltamethrin, and pirimiphos-methyl did not prevent hatching of *T. granarium* eggs, except pyrimiphos-methyl in the absence of food, they were able to kill all the emerged larvae. However, the addition of food slowed down this process, while no 100% mortality was recorded in treatments with pyriproxyfen and *s*-methoprene. Similarly, Saglam et al. [54] reported that spinetoram, thiamethoxam, and chlorantraniliprole were more effective against young and old larvae of the confused flour beetle, *Tribolium confusum* Jacquelin du Val (Coleoptera: Tenebrionidae), when applied on concrete surfaces where there was no food. As a mobile biological stage, larvae may mechanically remove the insecticides that are attached to their bodies through flour [43,55]. 

In a previous study, Kavallieratos et al. [27] found that chlorfenapyr, at label doses, caused 76.6% mortality of *T. granarium* small larvae after 7 days of exposure on treated concrete, while deltamethrin and pirimiphos-methyl caused <53% mortality at label doses. Furthermore, Athanassiou et al. [26] showed that α-cypermethrin killed 64.4% of small larvae of *T. granarium* on concrete, after the same exposure interval. In contrast, in our study, almost all larvae were dead in treatments with chlorfenapyr and deltamethrin, 7 days postexposure; while pirimiphos-methyl and α-cypermethrin caused 100% mortality after 5 days and 6 days of exposure, respectively. Our results indicate that when *T. granarium* is exposed to toxicants at the egg stage on treated surfaces, e.g., through aerial transfer during cleaning storage facilities [30], it is more susceptible to insecticides than when it is exposed at an early larval stage. This phenomenon was more vigorously expressed in the case of pyriproxyfen. Kavallieratos et al. [27] reported that pyriproxyfen killed <3.5% of small larvae on treated concrete, 7 days postexposure vs. 68% at the same exposure interval in the current study. The activity of JHAs gradually develops since it is related to the metamorphosis of larvae [56,57], indicating that larvae which survive treatments with pyriproxyfen and *s*-methoprene may be affected in the adult stage, though this issue was not investigated in the current study.

## 5. Conclusions

Overall, our study revealed that pirimiphos-methyl and α-cypermethrin were the most effective insecticides against *T. granarium*, causing high ovicidal activity and larval mortality; while, although chlorfenapyr and deltamethrin had lower ovicidal activity, they killed all the emerged larvae. The timing of insecticidal application is crucial for the adequate control of *T. granarium* given that exposure of eggs to insecticides increases the probability of death in this species at this stage or later in the larval stage. Food availability moderated and delayed the effect of almost all the insecticides tested. Therefore, it is important to keep storage facilities clean prior to the application of insecticides as it has also been suggested by previous studies [40,42,44,58,59,60]. To effectively manage this pest, more studies are required to examine the impact of additional contact insecticides on eggs of *T. granarium* when they are attached onto different grain kernels in different temperatures and relative humidity combinations.

## Figures and Tables

**Table 1 insects-11-00263-t001:** MANOVA parameters for the main effects and associated interactions for egg hatching and larval mortality of *Trogoderma granarium* (egg hatching total Degrees of Freedom, DF = 48, larval mortality with food DF = 47, larval mortality without food DF = 35).

Effect	Egg Hatching						Larval Mortality					
All Between	With Food			Without Food			With Food			Without Food		
	DF	*F*	*p*	DF	*F*	*p*	DF	*F*	*p*	DF	*F*	*p*
Intercept	1	759.5	<0.01	1	638.4	<0.01	1	4355.0	<0.01	1	4234.3	<0.01
Insecticide	5	18.3	<0.01	5	59.4	<0.01	5	23.2	<0.01	4	7.1	0.01
**All Within Interaction**												
Exposure	13	53.8	<0.01	13	38.3	<0.01	13	541.5	<0.01	13	320.1	<0.01
Exposure x insecticide	65	1.9	0.01	65	2.2	<0.01	65	2.2	<0.01	52	2.3	0.01

**Table 2 insects-11-00263-t002:** Mean percent (% ± SE) of egg hatching of *Trogoderma granarium* on concrete treated with α-cypermethrin, chlorfenapyr, deltamethrin, pirimiphos-methyl, pyriproxyfen, and *s*-methoprene, and on untreated concrete (control) after 1, 2, 3, 4, 5, 6, 7, 8, 9, 10, 11, 12, 13, and 14 days with or without food. Within each column, means followed by the same uppercase letter are not significantly different (in all cases DF = 13, 125, *p* < 0.01, Tukey–Kramer honest significant difference (HSD) test at 0.05). Within each row, means followed by the same lowercase letter are not significantly different (in all cases DF = 6, 62, *p* < 0.01, Tukey–Kramer HSD test at 0.05). Where no letters exist, no significant differences were recorded. Where dashes exist, no statistical analysis was performed.

Treatment	Control	α-Cypermethrin	Chlorfenapyr	Deltamethrin	Pirimiphos-methyl	Pyriproxyfen	*s*-Methoprene	*F*
	**With Food**							
Exposure								
1 day	0.0 ± 0.0 D	0.0 ± 0.0 C	0.0 ± 0.0 D	0.0 ± 0.0 C	0.0 ± 0.0 C	0.0 ± 0.0 B	0.0 ± 0.0 B	-
2 days	0.0 ± 0.0 D	0.0 ± 0.0 C	0.0 ± 0.0 D	0.0 ± 0.0 C	0.0 ± 0.0 C	0.0 ± 0.0 B	0.0 ± 0.0 B	-
3 days	81.1 ± 2.0 C,a	3.3 ± 2.4 B,C,d	23.3 ± 6.7 C,b,c	8.9 ± 3.5 C,c,d	3.3 ± 1.7 B,C,d	24.8 ± 5.5 A,b	11.1 ± 3.1 B,b,c,d	29.3
4 days	93.3 ± 2.4 B,a	11.1 ± 3.1 A,B,e	52.2 ± 6.4 B,C,b,c	41.1 ± 5.9 B,c,d	17.8 ± 4.7 A,B,d,e	44.4 ± 7.5 A,b,c	70.0 ± 6.9 A,b	24.5
5 days	100.0 ± 0.0 A,a	22.2 ± 4.3 A,d	76.7 ± 6.5 A,B,b	63.3 ± 5.8 A,b,c	24.4 ± 5.0 A,d	47.8 ± 8.0 A,c,d	78.9 ± 6.3 A,b	27.9
6 days	100.0 ± 0.0 A,a	28.9 ± 5.4 A,d	82.2 ± 6.6 A,B,b	73.3 ± 4.4 A,b,c	26.7 ± 5.0 A,d	47.8 ± 8.0 A,c,d	83.3 ± 4.7 A,b	28.7
7 days	100.0 ± 0.0 A,a	28.9 ± 5.4 A,c	85.6 ± 6.7 A,a,b	76.7 ± 3.7 A,b	26.7 ± 5.0 A,c	47.8 ± 8.0 A,c	83.3 ± 4.7 A,b	30.0
8 days	100.0 ± 0.0 A,a	28.9 ± 5.4 A,c	87.7 ± 5.5 A,a,b	76.7 ± 3.7 A,b	26.7 ± 5.0 A,c	47.8 ± 8.0 A,c	84.4 ± 5.0 A,b	31.6
9 days	100.0 ± 0.0 A,a	28.9 ± 5.4 A,c	87.7 ± 5.5 A,a,b	76.7 ± 3.7 A,b	26.7 ± 5.0 A,c	47.8 ± 8.0 A,c	86.7 ± 5.3 A,a,b	31.7
10 days	100.0 ± 0.0 A,a	28.9 ± 5.4 A,c	87.7 ± 5.5 A,a,b	76.7 ± 3.7 A,b	26.7 ± 5.0 A,c	47.8 ± 8.0 A,c	86.7 ± 5.3 A,a,b	31.7
11 days	100.0 ± 0.0 A,a	28.9 ± 5.4 A,c	87.7 ± 5.5 A,a,b	76.7 ± 3.7 A,b	26.7 ± 5.0 A,c	50.0 ± 9.1 A,c	86.7 ± 5.3 A,a,b	31.7
12 days	100.0 ± 0.0 A,a	28.9 ± 5.4 A,c	87.7 ± 5.5 A,a,b	76.7 ± 3.7 A,b	26.7 ± 5.0 A,c	50.0 ± 9.1 A,c	86.7 ± 5.3 A,a,b	31.7
13 days	100.0 ± 0.0 A,a	28.9 ± 5.4 A,c	87.7 ± 5.5 A,a,b	76.7 ± 3.7 A,b	26.7 ± 5.0 A,c	50.0 ± 9.1 A,c	86.7 ± 5.3 A,a,b	31.7
14 days	100.0 ± 0.0 A,a	28.9 ± 5.4 A,c	87.7 ± 5.5 A,a,b	76.7 ± 3.7 A,b	26.7 ± 5.0 A,c	50.0 ± 9.1 A,c	86.7 ± 5.3 A,a,b	31.7
*F*	948.8	15.8	31.4	73.3	8.4	10.9	34.9	
	**Without Food**							*F*
1 day	0.0 ± 0.0 D	0.0 ± 0.0	0.0 ± 0.0 C	0.0 ± 0.0 C	0.0 ± 0.0	0.0 ± 0.0 B	0.0 ± 0.0 C	-
2 days	0.0 ± 0.0 D	0.0 ± 0.0	0.0 ± 0.0 C	0.0 ± 0.0 C	0.0 ± 0.0	0.0 ± 0.0 B	0.0 ± 0.0 C	-
3 days	72.2 ± 2.8 C,a	1.1 ± 1.1 d	31.1 ± 8.6 B,b,c	24.4 ± 3.8 B,c	0.0 ± 0.0 d	34.4 ± 5.6 A,b,c	48.9 ± 4.8 B,b	44.9
4 days	92.2 ± 2.2 B,a	4.4 ± 2.9 c	58.9 ± 7.7 A,B,b	43.3 ± 3.7 A,B,b	0.0 ± 0.0 c	40.0 ± 5.3 A,b	83.3 ± 5.3 A,a	67.4
5 days	98.9 ± 1.1 A,a	6.7 ± 2.9 d	75.6 ± 6.3 A,b	54.4 ± 6.5 A,b,c	0.0 ± 0.0 d	42.2 ± 6.6 A,c	91.1 ± 3.5 A,a	73.1
6 days	98.9 ± 1.1 A,a	6.7 ± 2.9 e	76.7 ± 6.5 A,b,c	62.2 ± 7.4 A,c,d	0.0 ± 0.0 e	42.2 ± 6.6 A,d	91.1 ± 3.5 A,a,b	73.7
7 days	98.9 ± 1.1 A,a	6.7 ± 2.9 e	76.7 ± 6.5 A,b,c	63.3 ± 7.1 A,c,d	0.0 ± 0.0 e	42.2 ± 6.6 A,d	91.1 ± 3.5 A,a,b	74.8
8 days	98.9 ± 1.1 A,a	6.7 ± 2.9 e	76.7 ± 6.5 A,b,c	63.3 ± 7.1 A,c,d	0.0 ± 0.0 e	42.2 ± 6.6 A,d	91.1 ± 3.5 A,a,b	74.8
9 days	98.9 ± 1.1 A,a	6.7 ± 2.9 e	76.7 ± 6.5 A,b,c	63.3 ± 7.1 A,c,d	0.0 ± 0.0 e	42.2 ± 6.6 A,d	91.1 ± 3.5 A,a,b	74.8
10 days	98.9 ± 1.1 A,a	6.7 ± 2.9 e	76.7 ± 6.5 A,b,c	63.3 ± 7.1 A,c,d	0.0 ± 0.0 e	42.2 ± 6.6 A,d	91.1 ± 3.5 A,a,b	74.8
11 days	98.9 ± 1.1 A,a	6.7 ± 2.9 e	76.7 ± 6.5 A,b,c	63.3 ± 7.1 A,c,d	0.0 ± 0.0 e	42.2 ± 6.6 A,d	91.1 ± 3.5 A,a,b	74.8
12 days	98.9 ± 1.1 A,a	6.7 ± 2.9 e	76.7 ± 6.5 A,b,c	63.3 ± 7.1 A,c,d	0.0 ± 0.0 e	42.2 ± 6.6 A,d	91.1 ± 3.5 A,a,b	74.8
13 days	98.9 ± 1.1 A,a	6.7 ± 2.9 e	76.7 ± 6.5 A,b,c	63.3 ± 7.1 A,c,d	0.0 ± 0.0 e	42.2 ± 6.6 A,d	91.1 ± 3.5 A,a,b	74.8
14 days	98.9 ± 1.1 A,a	6.7 ± 2.9 e	76.7 ± 6.5 A,b,c	63.3 ± 7.1 A,c,d	0.0 ± 0.0 e	42.2 ± 6.6 A,d	91.1 ± 3.5 A,a,b	-
*F*	248.9	1.2	24.3	24.3	-	16.0	50.6	

**Table 3 insects-11-00263-t003:** Mean mortality (% ± SE) of *Trogoderma granarium* larvae on concrete treated with α-cypermethrin, chlorfenapyr, deltamethrin, pirimiphos-methyl, pyriproxyfen, and *s*-methoprene, and on untreated concrete (control) after 1, 2, 3, 4, 5, 6, 7, 8, 9, 10, 11, 12, 13, and 14 days with or without food. Within each column, means followed by the same uppercase letter are not significantly different (in all cases DF = 13, 125, except in the cases of pirimiphos-methyl with food DF = 13, 111 and α-cypermethrin without food DF = 13, 55, Tukey–Kramer HSD test at 0.05). Within each row, means followed by the same lowercase letter are not significantly different (in cases with food DF = 6, 61, in cases without food DF = 5, 48, Tukey–Kramer HSD test at 0.05). Where no letters exist, no significant differences were recorded. Where dashes exist, no statistical analysis was performed.

Treatment	Control	α-Cypermethrin	Chlorfenapyr	Deltamethrin	Pirimiphos-Methyl	Pyriproxyfen	*s*-Methoprene	*F*	*p*
	**With Food**		
Exposure									
1 day	0.0 ± 0.0	0.0 ± 0.0 C	0.0 ± 0.0 F	0.0 ± 0.0 D	0.0 ± 0.0 D	0.0 ± 0.0 E	0.0 ± 0.0 E	-	-
2 days	0.0 ± 0.0	18.5 ± 12.6 B,C,a	15.4 ± 4.4 E,a	2.6 ± 1.7 D,a,b	11.9 ± 6.5 D,a	0.0 ± 0.0 E,b	1.1 ± 1.1 E,b	2.5	0.03
3 days	0.0 ± 0.0	35.2 ± 13.4 B,a,b	62.3 ± 7.1 D,a	44.6 ± 9.0 C,a	37.7 ± 9.3 C,a,b	0.0 ± 0.0 E,c	5.0 ± 2.0 D,E,b,c	11.6	<0.01
4 days	0.0 ± 0.0 b	79.9 ± 11.1 A,a	76.6 ± 4.7 C,D,a	71.1 ± 4.7 B,a	75.6 ± 8.2 B,a	14.9 ± 7.4 D,E,b,c	24.7 ± 5.4 C,D,b	21.6	<0.01
5 days	0.0 ± 0.0 c	98.2 ± 5.6 A,a	85.7 ± 3.1 B,C,b	96.8 ± 2.1 A,a,b	100.0 ± 0.0 A,a	41.0 ± 7.6 C,D,c	49.3 ± 8.0 B,C,c	74.4	<0.01
6 days	0.0 ± 0.0 c	100.0 ± 0.0 A,a	93.2 ± 2.4 A,B,a	96.8 ± 2.1 A,a	100.0 ± 0.0 A,a	56.2 ± 9.8 B,C,b	62.5 ± 7.0 A,B,b	60.3	<0.01
7 days	0.0 ± 0.0 d	100.0 ± 0.0 A,a	98.8 ± 1.2 A,a	96.8 ± 2.1 A,a	100.0 ± 0.0 A,a	68.0 ± 10.0 A,B,C,b	72.8 ± 7.1 A,B,b	55.9	<0.01
8 days	0.0 ± 0.0 c	100.0 ± 0.0 A,a	100.0 ± 0.0 A,a	100.0 ± 0.0 A,a	100.0 ± 0.0 A,a	72.0 ± 10.6 A,B,C,b	77.2 ± 6.7 A,B,b	60.5	<0.01
9 days	0.0 ± 0.0 c	100.0 ± 0.0 A,a	100.0 ± 0.0 A,a	100.0 ± 0.0 A,a	100.0 ± 0.0 A,a	83.1 ± 5.9 A,B,b	80.5 ± 6.0 A,B,b	113.6	<0.01
10 days	0.0 ± 0.0 c	100.0 ± 0.0 A,a	100.0 ± 0.0 A,a	100.0 ± 0.0 A,a	100.0 ± 0.0 A,a	88.8 ± 4.6 A,b	84.6 ± 4.9 A,b	148.6	<0.01
11 days	0.0 ± 0.0 c	100.0 ± 0.0 A,a	100.0 ± 0.0 A,a	100.0 ± 0.0 A,a	100.0 ± 0.0 A,a	91.6 ± 4.4 A,a,b	85.7 ± 5.2 A,b	143.2	<0.01
12 days	0.0 ± 0.0 c	100.0 ± 0.0 A,a	100.0 ± 0.0 A,a	100.0 ± 0.0 A,a	100.0 ± 0.0 A,a	91.6 ± 4.4 A,a,b	85.7 ± 5.2 A,b	143.2	<0.01
13 days	0.0 ± 0.0 c	100.0 ± 0.0 A,a	100.0 ± 0.0 A,a	100.0 ± 0.0 A,a	100.0 ± 0.0 A,a	92.8 ± 3.6 A,a,b	87.3 ± 4.2 A,b	173.5	<0.01
14 days	0.0 ± 0.0 c	100.0 ± 0.0 A,a	100.0 ± 0.0 A,a	100.0 ± 0.0 A,a	100.0 ± 0.0 A,a	94.1 ±3.1 A,a,b	87.3 ± 4.2 A,b	189.9	<0.01
*F*	-	37.9	138.3	140.8	97.7	32.1	34.7	-	-
*p*	-	<0.01	<0.01	<0.01	<0.01	<0.01	<0.01		
	**Without Food**	*F*	*p*
1 day	0.0 ± 0.0	0.0 ± 0.0 B	3.6 ± 2.6 D	5.7 ± 3.0 E	-	0.0 ± 0.0 D	0.0 ± 0.0 F	2.2	0.07
2 days	0.0 ± 0.0 b	12.5 ± 12.5 B,a,b	36.9 ± 10.8 C,a	31.2 ± 6.8 D,a	-	3.7 ± 3.7 D,b	10.4 ± 3.4 E,a,b	7.0	<0.01
3 days	0.0 ± 0.0 c	75.0 ± 25.0 A,a	82.9 ± 5.8 B,a	61.0 ± 7.1 C,a,b	-	49.8 ± 13.7 C,a,b	16.7 ± 3.3 D,e,b,c	12.0	<0.01
4 days	0.0 ± 0.0 d	100.0 ± 0.0 A,a	95.3 ± 2.4 A,B,a	89.8 ± 3.0 B,a,b	-	60.9 ± 10.9 B,C,b	36.0 ± 6.5 D,c	40.6	<0.01
5 days	0.0 ± 0.0 c	100.0 ± 0.0 A,a	100.0 ± 0.0 A,a	96.6 ± 2.4 A,B,a	-	78.7 ± 10.9 A,B,C,a	62.1 ± 5.3 C,b	59.6	<0.01
6 days	0.0 ± 0.0 c	100.0 ± 0.0 A,a	100.0 ± 0.0 A,a	97.8 ± 2.2 A,b,a,b	-	83.3 ± 8.3 A,B,C,a,b	82.6 ± 7.1 B,b	60.4	<0.01
7 days	0.0 ± 0.0 b	100.0 ± 0.0 A,a	100.0 ± 0.0 A,a	100.0 ± 0.0 A,a	-	86.1 ± 7.4 A,B,a	95.0 ± 2.7 A,B,a	106.7	<0.01
8 days	0.0 ± 0.0 c	100.0 ± 0.0 A,a	100.0 ± 0.0 A,a	100.0 ± 0.0 A,a	-	88.0 ± 6.4 A,B,b	100.0 ± 0.0 A,a	173.8	<0.01
9 days	0.0 ± 0.0 b	100.0 ± 0.0 A,a	100.0 ± 0.0 A,a	100.0 ± 0.0 A,a	-	95.4 ± 3.1 A,B,a	100.0 ± 0.0 A,a	431.3	<0.01
10 days	0.0 ± 0.0 b	100.0 ± 0.0 A,a	100.0 ± 0.0 A,a	100.0 ± 0.0 A,a	-	95.4 ± 3.1 A,B,a	100.0 ± 0.0 A,a	431.3	<0.01
11 days	0.0 ± 0.0 b	100.0 ± 0.0 A,a	100.0 ± 0.0 A,a	100.0 ± 0.0 A,a	-	95.4 ± 3.1 A,B,a	100.0 ± 0.0 A,a	431.3	<0.01
12 days	0.0 ± 0.0	100.0 ± 0.0 A	100.0 ± 0.0 A	100.0 ± 0.0 A	-	100.0 ± 0.0 A	100.0 ± 0.0 A	-	-
13 days	0.0 ± 0.0	100.0 ± 0.0 A	100.0 ± 0.0 A	100.0 ± 0.0 A	-	100.0 ± 0.0 A	100.0 ± 0.0 A	-	-
14 days	0.0 ± 0.0	100.0 ± 0.0 A	100.0 ± 0.0 A	100.0 ± 0.0 A	-	100.0 ± 0.0 A	100.0 ± 0.0 A	-	-
*F*	-	20.8	75.1	85.6	-	24.8	130.8		
*p*	-	<0.01	<0.01	<0.01	-	<0.01	<0.01

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
