# Peer review of "Effect of Six Insecticides on Egg Hatching and Larval Mortality of Trogoderma granarium Everts (Coleoptera: Dermestidae)"

_insects, 2020, doi:10.3390/insects11050263_

Round 1
Reviewer 1 Report
For the most part, this paper is fairly well-written and will require only minor revision. Specific comments for the authors are given below.
Abstract-Numerous spelling errors, inconsistencies in wording, etc.
line 14, where are these insecticides registered. You need a specific country, all countries vary in what is or isn’t registered for control of stored product insects on interior surfaces. Otherwise delete the phrase.
Line 15, modes of action, not mode of actions Line 16, you don’t need
“of this species”
Line 16, on treated concrete not in
Line 19, again, it is on treated concrete. Check the entire paper and
use phrase on treated concrete Line 21, now you are calling them
treated dishes not treated concrete Line 29, now you say on treated
concrete
Introduction
Lines 41-43, T. granarium is subject to regulations in most developed countries, you don’t need to list individual countries as examples.
Line 45, body parts not parts of it’s body Line 47-48, poor English
grammar. Use Kav. showed that T. granarium rapidly reproduces on
animal products (delete several unless you define them) Line 55,
insert for before up Line 57, combine this short two-sentence paragraph with the previous paragraph.
Materials and Methods.
Combine the first three sections, they are too short.
Results
Explains the Tables fairly well, only one comment Line 143, Combine
this one-sentence paragraph with the next one.
Discussion
Well-written, just one comment
Line 244, food not feed
Tables
The Font size differs in all Tables. I realize there are a lot of data
in Tables 2 and 3, Table 1-match font size of Table 3.
Table 2-P values are < 0.01 in all cases. Delete the column, put this info in the Table Heading, and enlarge the font size to match Table 3. Delete the p values in the row too, they are all the same, put the info in the Table Heading or in a Footnote.
Reviewer 2 Report
The authors detail a study examining the effects of 6 contact insecticides on the hatching and larval survival of khapra beetle. They authors examined the effect of the insecticides on concrete surfaces with and without food. The insecticides show varying success in egg hatching and larval mortality. Food typically decreased the effect of the insecticide, although not in all cases.
Specific comments:
Line 90: Extra “and” in your list?
Line 109: Was the flour in a pile or sprinkled over the surface area of the concrete?
Methods: Were all eggs tested at once? Or were there blocks within this design? Also I assume that it is 10 replicate dishes (eggs) per insecticide per food treatment? You should say for sure what your level of replication is.
Line 134-135: Is food your repeated measure? Not clear from current phrasing.
Line 143: Where does this exposure interval come from? It is not detailed in the methods section or mentioned in your model description. If your 14 day observation is this factor, you need to say this in the methods.
Line 157-165: Why is there such a difference among insecticides in the presence of food on egg hatching? What do you know about these insecticides that could drive this? Please discuss more in the discussion section as this is important to drawing general conclusions.
Line 167: Maybe you can reiterate here the actual numbers of larvae that you examined for mortality for each treatment. That would give a better idea of the numbers you are comparing in your model.
Results Overall: The results are a little bit confusing. It seems like it is a lot of lists of numbers and percentages and it is hard to keep everything straight. Please just take a deep read and see if you can reorganize or provide a graph or two for some visual representation of egg hatching and mortality.
Line 168: “Without” is missing.
Line 213: Formatting issue
Line 220: Greater instead of greated?
Discussion: I think the overall results of the manuscript are pretty straight forward but there are still some questions regarding the use of these insecticides. What are the differences among the insecticides and why did you need to test them all? Maybe that is something for the introduction and then you can explain the knowledge that applicators can use from these data.
